# The Color Formation Mechanism of the Blue Karst Lakes in Jiuzhaigou Nature Reserve, Sichuan, China

**Xiaohui Li [1,2], Mengqi Zhang [1], Weiyang Xiao [3], Jie Du [3], Meiqun Sheng [1], Dalin Zhu [1], Anđelka Plenković-Moraj [4] and Geng Sun [1,*]**

[1] CAS Key Laboratory of Mountain Ecological Restoration and Bioresource Utilization & Ecological Restoration and Biodiversity, Conservation Key Laboratory of Sichuan Province & China-Croatia "Belt and Road" Joint Laboratory on Biodiversity and Ecosystem Services, Chengdu Institute of Biology, Chinese Academy of Sciences, Chengdu 610041, China; lixh@cib.ac.cn

[2] University of Chinese Academy of Sciences, Beijing 100049, China

[3] Jiuzhaigou Nature Reserve Administrative Bureau, Zhangzha Town, Jiuzhaigou County, Sichuan 623402, China

[4] Department of Biology, Faculty of Science, University of Zagreb, Rooseveltov Trg 6, 10000 Zagreb, Croatia; aplenk@biol.pmf.hr

[*] Correspondence: sungeng@cib.ac.cn; Tel.: +86-28-8289-0614

**Abstract:** The karst lakes in Jiuzhaigou Nature Reserve are an integral part of the karst lake landscape, yet research on the formation mechanism for the color of the blue-green lakes in Jiuzhaigou is insufficient. With the help of hyperspectral instruments, coupled with hydro-chemical analysis, this paper elaborates on the unique color characteristics of the Jiuzhaigou karst lakes, delves into the color formation mechanism of the lakes, establishes a regression equation for the color of the lakes as well as the water quality parameters, and sheds light upon the causes for the color distinction between the karst lakes in Jiuzhaigou and the plateau freshwater lakes. The experiment shows that the Jiuzhaigou karst lakes are primarily blue and green, while the proportion of short-wavelength light in the normalized water-leaving radiance and the total incident irradiance of lake water is higher. Based on the redundancy analysis and the correlation analysis, travertine deposition is the core link in the color formation of the blue karst lakes in Jiuzhaigou, while the selective reflection and scattering of the suspended calcium carbonate particulate matters towards visible light represents the optical foundation for the formation. In addition, physical factors such as depth and transparency, changes to the water quality parameters that affect the travertine deposition rate, and the eutrophication process will all exert significant influence over the formation. By building on water-leaving radiance, this paper quantifies the lake color with the tristimulus values (R, G, B) via colorimetrical methods, which features solid goodness of fit with the linear regression equation established based on the water quality parameters. The principal component analysis and colorimetrical analysis show that the color of the karst lakes in Jiuzhaigou varies substantially from that of the plateau freshwater lakes, which mainly results from the difference in the water quality. Research conducted in this paper on the color formation mechanism of the distinct blue karst lakes in Jiuzhaigou illuminates the formation and maintenance mechanism of the plateau karst lakes, which is conducive to better understanding towards the relationship between the water quality and colors of the karst lakes, and provides scientific proof for the establishment of the water quality assessment indicator system based on the colors of the karst lakes.

**Keywords:** Jiuzhaigou; karst lakes; blue-green lakes; travertine deposition; water quality

## 1. Introduction

Karst is a type of special landscape formed by chemical corrosion, erosion, and collapse on soluble rocks such as carbonate rocks, gypsum, marble, quartzite, etc. [1]. Karst lakes are one of the various karst landscapes that widely exist worldwide, and a large number of distinctive natural landscapes have been nurtured on that basis. For instance, Jiuzhaigou Nature Reserve [2–4], Huanglong Scenic Area [5], and the Plitvice Lakes National Park [6,7], Mammoth Cave National Park [8], etc. The essence of karst lakes lies in the colors. Thanks to the unique geographical conditions and ecological environment, most of the karst lakes boast clear and highly transparent water [9], whose colors are primarily bright blue-green and multi-layered. Renowned for its picturesque natural landscape, Jiuzhaigou Nature Reserve is an important karst scenic area in South China Karst, attracting tens of thousands of domestic and foreign tourists every year [10]. Inside the area scatter various karst lakes, whose crystal clear water, in combination with the travertine sediments and algae at the bottom, gives rise to a distinctive blue-green color lake landscape [11]. However, there is limited research on the reasons for the formation of the blue color of those lakes and the potential influence of water quality changes over the colors. Throughout the world, there are numerous karst lakes, most of which are blue-green. It is generally believed that colors of the karst lakes are closely intertwined with the calcium carbonate particulate matters in the water, whose scattering of light can make the lake water turquoise-colored [12,13]. For example, the Havasu Falls and the surrounding pools in the American Grand Canyon are abundant in calcium carbonate particulate matters due to noticeable travertine deposition, and their colors are mainly bright blue-green [14]. Color formation of the Bear Lake in Utah, the United States is relevant with calcium carbonate particulate matters as well, the mainstay of whose geological composition is limestone deposits. Weathering enabled a great amount of calcium carbonate to enter the lake, and the reflection towards blue light by the suspended calcium carbonate particulate matters in high concentrations in the lake made it sky-blue [15,16]. Color of the Blue Lake on Mount Gambier in South Australia alters with the season. Owing to the absorption of incident light by humus, the lake water is greyish blue in winter. In summer, there is coprecipitation between the calcite crystal formed by carbon dioxide gas removal as well as temperature increase and humus, which has a regulating effect on the density of humus over the surface of the lake. On top of that, calcite crystal itself is conducive to the scattering of short-wavelengths, and the lake color will change from greyish blue to cobalt blue [17]. Due to the vulnerability of the karst lake aquatic ecosystem, the eutrophication process can also significantly change the watercolor. The input of nutrients and organisms led to the eutrophication in part of the karst lakes at the Mexican Lagunas de Montebello National Park, and the lake color changed from transparent blue to murky yellow-green consequently. In the degenerated lakes, concentrations of nutrient elements, chlorophyll-a, selenium, and particulate organic carbon are higher than those in the non-degenerated ones, and the particle sizes of the sediments are smaller. Because of the advanced underground water circulation system in the karst lakes, pollutants are likely to spread to the other lakes in the area [18]. Apart from karst lakes, many non-karst ones can also be blue with the suspended particulate matters in the water. Ohsawa et al. carried out a research on the color formation mechanism of part of the blue hot springs in Japan, New Zealand and Yunnan, China, in which they found the water was blue on account of the Rayleigh scattering by the water-soluble silicon dioxide particulate matters towards the sunshine. The size of the colloidal silica sol ranges from 0.1–0.45 µm, smaller than the wavelength in the visible spectrum [19]. Onda et al. discussed the color formation causes of the typical Japanese emerald volcanic lakes with high acidity via colorimetric analysis, hydro-chemical analysis and particle distribution data. The conclusion was that the $SO_2$ (aq) and $H_2S$ (aq) formed through the fumaroles at the lake bottoms gave rise to colloidal sulfur particulate matters, the Rayleigh scattering and Mie scattering by which towards sunshine brought into existence the distinctive lake color. Besides, the absorption of sunshine by the water-soluble ferrous ions made part of the water green [20]. Castellón et al. investigated the color formation causes of the blue Rı́o Celeste at the Costa Rican Tenorio National Park and found the convergence of two tributaries resulted in changes of the pH value of Rı́o Celeste, which further led to the agglomeration of colloidal silica aluminate particulate matters, with their sizes augmenting from 184

nm to 566 nm. The sub-micron cluster of particulate matters formed, therefore, exercise Mie scattering towards the sunshine, thus making the Rı́o Celeste blue. The white sediments formed after those particulate matters descended to the bottom of the river reinforced such a sky-blue landscape [21]. Some glacier lakes and rivers on high mountains are sky-blue owing to the glacier flour or rock dust existing in the water [22,23], the suspended matters in which are formed because of the grinding of sediment particulates in the glaciers during movements [24].

In recent years, a great deal of research has made it clear that changes to water quality will alter watercolor. Absorption and scattering by the suspended matters, plankton, colored dissolved organic matters (CDOM), and other optically active substances in the water towards incident ray will lead to reduced light and changes to the euphotic depth, which will ultimately alter the color of the lake. The rise of suspended matters in the water will have a direct impact on the transparency and elevate the reflectivity [25]. The selective absorption by planktonic algae of blue light will change the lake color into green or red [26,27]. As a type of important light absorption substance, CDOM has intense absorption in the ultraviolet band and the blue band, which can make the water yellow or brown [28,29].

Currently, the major watercolor measurement methods are the platinum-cobalt method [30] and the multiple dilution method [31], which are both based on judgments made with the naked eye and are therefore subjective. To make the results more accurate, many scholars deployed spectroscopic and colorimetrical methods to establish the correlation equations for water absorbance, chromaticity and chromatic aberration, and calculated the water chromaticity with absorbance data [32,33]. Nevertheless, the abovementioned measurement methods are implemented indoors, which fail to take into consideration the dissemination of visible light in the water and the reflection and scattering by water-soluble substances towards the light. The chromaticity obtained in this manner cannot represent the actual color seen with the naked eye. Over the past couple of years, hyperspectral remote sensing has been widely applied in the analysis of water quality as well as the research on colors. Some researchers set up a spectral color measurement system with the help of the SOC710VP hyperspectral imager, source of natural light and water tank, and explored the impact of various depths and suspended matter concentrations on natural landscape water through indoors simulation experiment. Moreover, they established the fitted equation between water spectrum reflectivity and suspended matter concentrations [34]. Oyama et al. measured the water-leaving radiance, watercolor factor absorption coefficient, scattering coefficient, and other innate optical features of water in the Onneto shallow lake in Hokkaido, Japan with ASD FieldSpec HandHeld. Through colorimetrical analysis and the establishment of a biological-optical model, the researchers delved into the formation causes of the lake color and found that the backscattering of long-wavelengths by suspended solids with the diameter over 0.7 μm is an indispensable influencing factor of the lake color [35]. Some other researchers also measured the seawater water-leaving radiance, remote sensing reflectance under different phytoplankton content, CDOM concentration, and suspended matter concentration by utilizing Hyperspectral Tethered Spectroradiometer Buoy (HTSRB, Satantic, Inc.) and HyperPro II (Satlantic Inc.). With a colorimetrical analysis and quantitative model, they shed light upon reasons why the seawater changes from blue to red, brown and black [27]. Distinctive colors are the core of the karst lake landscape, whereas there are few reports on the quantitative assessment methods as well as the mechanism for the color of the blue karst lakes in Jiuzhaigou Nature Reserve, and the role of water quality changes in the formation of the blue lake water is yet to be identified.

In order to comprehend the formation mechanism and influencing factors of the unique blue color of the karst lakes in Jiuzhaigou, we set up water sampling points at Jiuzhaigou karst lakes such as Colorful Lake, Mirror Lake as well as Rhinoceros Lake, and the plateau freshwater lakes including Cuoqiong Lake and Xiarucuo Lake. During the wet season of the lakes in the summer of 2019, we collected the water-leaving radiance ($L_W$) and the incident irradiance ($E_d$) data of the lake at the said sampling points with ASD FieldSpec HandHeld 2 and collected water samples to carry out indoors water quality analysis. For the first time, we described the unique color traits of the Jiuzhaigou karst lakes, probed into the color formation mechanism as well as major influencing factors of water quality

for the blue karst lakes, established the regression equation between watercolors and water quality factors, and expounded on the causes for the color differences between the Jiuzhaigou karst lakes and the plateau freshwater lakes. Research conducted in this paper will not only be helpful to the understanding of the formation and maintenance mechanism for the distinctive colors of the general plateau karst lakes but also provide scientific proof for the establishment of the water quality assessment indicator system based on the karst lake colors.

## 2. Materials and Methods

### 2.1. Study Area

Jiuzhaigou Natural Reserve is at the southern part of the Minshan Mountains on the eastern edge of the Tibetan Plateau. It is situated between 103°46′-104°50′ E and 32°55′-33°20′ N. With the altitude ranging from 1996 m to 4764 m, it is administered by the Jiuzhaigou County, Aba Tibetan and Qiang Autonomous Prefecture, Sichuan Province [36,37]. Covering a land area of 720 km² and a water area of 2.85 km², the reserve is at a trench along the valley with the depth of approximately 40 km, chief of which are branches including the Rize Valley, the Zechawa Valley and the Shuzheng Valley (Figure 1a).Water in the valleys mainly comes from precipitation, mountain meltwater and underground karst water. Travertine is a type of calcium carbonate deposit [38] and also an important part of the landscape in Jiuzhaigou. Primarily distributed in the Rize Valley, the Shuzheng Valley and their branches, it stretches for around 30 km intermittently and covers 2.44 km². Featuring diverse types and unique shapes, it is rarely seen throughout the globe [39]. Endowed with distinctive alpine karst landform, under special geological and hydrological conditions, Jiuzhaigou Natural Reserve is home to geological relics such as karst lakes, calcareous sinter flows and calcareous falls. The majority of the lakes in the Valley are oligotrophic ones. Due to karstification, there is a great quantity of travertine deposit at the bottom of the lakes and the lake water appears a unique blue green [40].

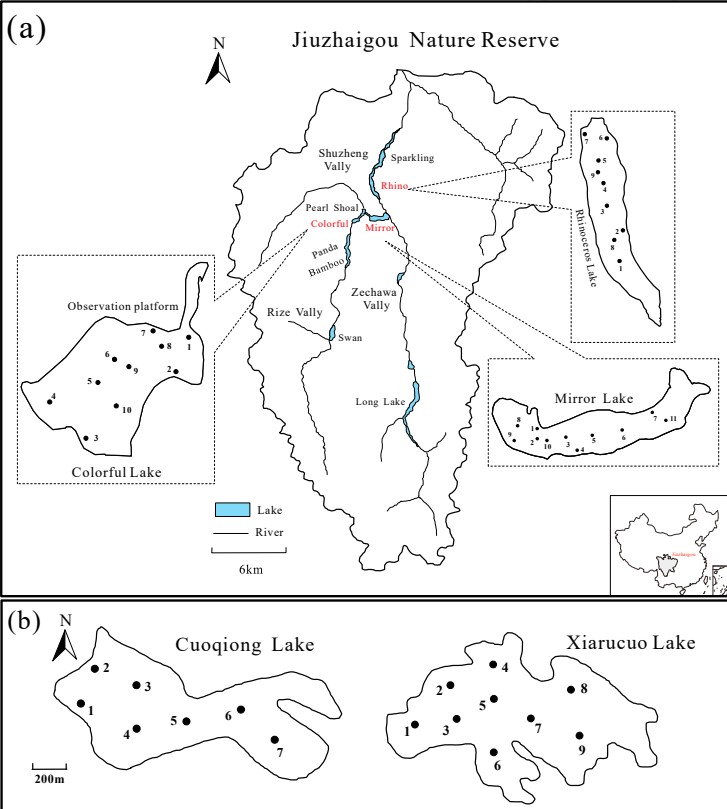

**Figure 1.** Maps showing: (**a**) Geographical location of the Jiuzhaigou Natural Reserve and sampling sites in different lakes; (**b**) Sampling sites in Cuoqiong Lake and Xiarucuo Lake.

*2.2. Sampling*

Following the actual sampling needs, 30 monitoring points were set up at the three representative karst lakes of Colorful Lake, Mirror Lake and Rhinoceros Lake in Jiuzhaigou, and 16 points at the plateau freshwater lakes of Cuoqiong Lake and Xiarucuo Lake. Points nearby each lake covered the entire lake area, including the inflow area, the outflow area, the lake center, the deep water area, the shallow water area, and the area nearby the observation platform (Figure 1). Spectral data collection and water sampling were carried out simultaneously during the sunny days in July and August 2019. An ASD FieldSpec HandHeld 2 was used to collect in situ the water-leaving radiance ($L_W$) and incident irradiance ($E_d$) data. At the same time, in total 254 bottles of water samples were collected with plexiglass hydrophore at different layers in the lakes. During the sampling process, if the lake was less than 5 m deep, then the sampling points would be set at 0.5 m under the surface. If it was more than 5 m, then the points would be set at 0.5 m under the surface and 0.5 m away from the bottom. As the hydrological conditions of the karst lakes are special, substantial water quality differences might occur on the same surface. Therefore, multiple points would be added in the middle per depths of the lakes. The water samples collected were kept in a portable refrigerator at the temperature of 4℃, then transported to the laboratory in Chengdu for analyses within 3 days after collection. Before sampling, the water temperature (WT), pH, total dissolved solids (TDS), conductivity, salinity, and dissolved oxygen (DO) were measured at each site by using a portable multi-parameter probe (Hach-sensION 156, US). The Secchi depth (SD) was determined by a standard 30-cm-diameter Secchi disk.

*2.3. Measurement of Water Quality Parameters*

The water quality parameters include perceptible physical water quality indicators such as: WT, chromaticity, turbidity (Turb) and transparency (SD); other indicators: conductivity, dissolved oxygen (DO), total dissolved solids (TDS); chemical indicators: pH, salinity, $K^+$, $Ca^{2+}$, $Na^+$, $Mg^{2+}$, $HCO_3^-$, $F^-$, $Cl^-$, $NO_3^-$, $SO_4^{2-}$; water eutrophication indicators: total nitrogen (TN), total phosphorus (TP), ratio of nitrogen to phosphorous (N/P), and colored dissolved organic matters (CDOM). Water temperature was measured with Hach-SensIon 156; chromaticity with spectrophotometry method [41]; turbidity with UV spectrophotometer [42]; transparency with a standard 30-cm-diameter Secchi disk [43]; conductivity, DO, TDS, pH, and salinity with Hach-SensIon 156; concentrations of cation and anion in the water with Metrohm 883, Swiss; TN with the total organic carbon (TOC) analyzing instruments manufactured by the German Elementar; TP with inductively coupled plasma emission spectroscopy (ICP-AES) by the American Thermo Jarrell Ash; concentrations of CDOM are demonstrated by water absorption coefficient at 440 nm, which can be obtained via the absorption coefficient equation [44,45].

*2.4. In Situ Collection of Water-Leaving Radiance and Incident Irradiance Data*

To describe the unique color of the Jiuzhaigou karst lakes, the normalized water-leaving radiance ($L_{WN}$) and water surface incident irradiance ($E_d$) at the sampling points are required. The former refers to the radiance emitted after sunshine enters water via the water-gas interface and becomes affected by the absorption as well as scattering by components in the water. It is a type of apparent optical property that often changes with the external light field [46]. An ASD Fieldspec HandHeld 2 was used to obtain water-leaving radiance and water surface incident irradiance with a spectral range from 325 nm to 1075 nm, following the Ocean Optics Protocols for Satellite Ocean Color Validation published by NASA [47]. As shown in Figure 2, the angle between the instrument observation plane and the solar incident plane is $90° \leq \phi \leq 135°$, while that between the instrument and the normal is $30° \leq \theta \leq 45°$. When observation was conducted with the instrument facing the water, the former was rotated upward at the observation plane so that the zenith angle of skylight spectral radiance $L_{sky}(\theta, \phi, \lambda)$ was in sync with the water observation angle ($\theta$) in avoidance of impact by perpendicular incidence as well as reflection of sunshine and shadows cast by ships on the results [48,49].

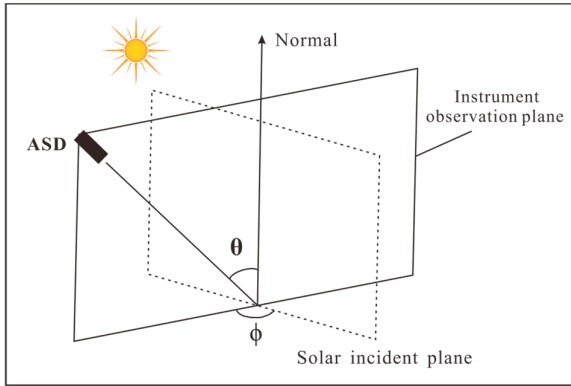

**Figure 2.** Water Spectral Observation Geometrics.

*2.5. Calculation of Normalized Water-Leaving Radiance and Incident Irradiance*

To minimize the measurement errors made by instruments, intervals between the measurement times should be shortened to the greatest extent. Five parallel measurements of the spectral data were carried out at each sampling point and the average was taken as final. The equation for the water-leaving radiance is as follows:

$$L_W(\theta, \phi, \lambda) = L_u(\theta, \phi, \lambda) - \rho L_{sky}(\theta, \phi, \lambda) \tag{1}$$

Where $L_W(\theta, \phi, \lambda)$ represents water-leaving radiance; $L_{sky}(\theta, \phi, \lambda)$ is skylight spectral radiance; $\rho$ is reflectivity of the gas-water interface towards skylight, which generally ranges between 2.1%–5% depending on location of the sun, light field distribution of skylight, observation geometrics, wind speed, wind direction, and other factors [50].

Nonetheless, water-leaving radiance changes with the lighting conditions. Only by normalizing it can the values obtained at different times and locations under different atmospheric conditions be comparable. Normalization indicates making sure the sun is right above the location of measurement to reduce impact by the atmosphere [51]. Definition of the normalized water-leaving radiance is:

$$L_{WN} = \frac{F_0}{E_d(\lambda)} L_W \tag{2}$$

$F_0$ stands for the solar radiation accepted per unit area per second where there is perpendicular to the sunshine outside the atmosphere, also known as the solar constant. This paper takes it as 1367 W/m² [52]; $E_d(\lambda)$ represents the water surface incident radiance, which can be calculated with the reflection intensity of standard whiteboard ($L_P$) measured:

$$E_d(\lambda) = L_P \pi / \rho_P \tag{3}$$

Where $L_P$ is the reflection intensity of standard whiteboard directly measured with instruments; $\rho_P$ is the reflectivity of the standard whiteboard. The reflectivity of different standard whiteboards varies, and that of the one deployed in this experiment can be found in the instructions.

*2.6. Quantitative Assessment of the Lake Color*

According to colorimetry, within the scope of visible light, for all colored objects in nature, once the spectral feature distribution that simulates the human eyes is obtained, their tristimulus values (X, Y, Z) can be calculated. The water-leaving radiance carries the spectral information related to the water. While people observe the color of the lake, the water-leaving radiance is the spectral feature

distribution that stimulates human eyes [53]. Through the equation below, colors observed with human eyes at the sampling points of each lake can be deducted. The perceived color assumes that the individual is looking directly down at a relatively calm water surface with no contrasting skylight or water surface glint. [27].

$$X = K \int_{380}^{780} L_w(\lambda)\, \bar{x}(\lambda) d\lambda$$

$$Y = K \int_{380}^{780} L_w(\lambda)\, \bar{y}(\lambda) d\lambda \qquad (4)$$

$$Z = K \int_{380}^{780} L_w(\lambda)\, \bar{z}(\lambda) d\lambda$$

$$K = \frac{100}{\int_{380}^{780} L_w(\lambda)\bar{y}(\lambda) d\lambda} \qquad (5)$$

Where X, Y and Z are the tristimulus values [54,55], which represent the quantities of the three primary colors required to match the color stimulus to be measured. X is red, Y green and Z blue; $\bar{x}(\lambda)$、 $\bar{y}(\lambda)$ and $\bar{z}(\lambda)$ are the spectral three irritation values of the 1931CIE chromaticity system, which can be obtained by the three-stimulus value table of the CIE standard chromaticity observation [34]; K is the adjustment coefficient.

RGB values of the lake colors can be obtained via equations (6)-(9):

$$X' = X/100$$

$$Y' = Y/100 \qquad (6)$$

$$Z' = Z/100$$

$$\begin{bmatrix} R' \\ G' \\ B' \end{bmatrix} = \begin{bmatrix} 3.240479 & -1.537150 & -0.498535 \\ -0.969256 & 1.875992 & 0.041556 \\ 0.055648 & -0.204043 & 1.057311 \end{bmatrix} \times \begin{bmatrix} X' \\ Y' \\ Z' \end{bmatrix} \qquad (7)$$

$$\text{Var-R} = R'^{\,0.455}$$

$$\text{Var-G} = G'^{\,0.455} \qquad (8)$$

$$\text{Var-G} = G'^{\,0.455}$$

$$R = \text{Var-R} \times 255$$

$$G = \text{Var-G} \times 255 \qquad (9)$$

$$B = \text{Var-B} \times 255$$

The calculation can be conducted with the MATLAB code. The resulting RGB color values range from 0 to 255. Values calculated to be less than 0 were set to 0 and values greater than 255 were set to 255. Once the RGB color values are obtained, watercolors can be simulated, and the regression equation can be established with the water quality parameters.

### 2.7. Data Analysis

To describe the color features of the Jiuzhaigou blue karst lakes, we first calculated the normalized water-leaving radiance and incident irradiance at the sampling points of each lake with the original spectral data based on equations (1)–(3), drew a spectral graph of the Jiuzhaigou karst

lakes and the plateau freshwater lakes within the scope of the visible light, and analyzed the primary characteristics of the spectral curves. Next, we took 630–780 nm, 500–570 nm and 420–470 nm as the characteristic wavelengths of red, green and blue light to calculate the normalized water-leaving radiance $L_{WN}$ (R), $L_{WN}$ (G) and $L_{WN}$ (B) and incident irradiance $E_d$ (R), $E_d$ (G) and $E_d$ (B), and analyzed the incident light as well as reflected light spectral distribution characteristics of the karst and freshwater lakes. Having specified the unique color characteristics of the karst lakes, we took the average of the water quality data measured at different layers of various sampling points as values for the lake. The calcite saturation index ($SI_C$) can be obtained via the earth chemical simulation software PHREEQC [56,57]. The equation is as follows:

$$SI_C = \lg\frac{I_{AP}}{K_C} = \lg\frac{[Ca^{2+}]\,[CO_3^{2-}]}{K_C} \tag{10}$$

Where $SI_C$ denotes the calcite saturation index, and $I_{AP}$ is the ion activity of a certain solid in the water. $K_C$ stands for the equilibrium constant of calcite under a certain temperature. When $SI_C = 0$, the chemical reaction system in the water is in the equilibrium state, while the dissolution and deposition speeds of the calcite are equal; when $SI_C < 0$, the chemical reaction system is unsaturated, and dissolution of the calcite is faster than its deposition; when $SI_C > 0$, the chemical reaction system is super-saturated and the calcite generates deposits [5].

Based on the water-leaving radiance data measured in real-time, the tristimulus values (R, G, B) collected at the sampling points were calculated via equation (4)–(9), and the lake colors were quantified. To investigate the relationship between colors and water quality of the Jiuzhaigou karst lakes, we carried out redundancy analysis and correlation analysis ($p = 0.05$) on the water quality as well as R, G, B data of the sampling points. In so doing, we explored the water quality indicators correlated with the blue color of the karst lakes and conducted stepwise regression analysis with them on the R, G, B values to verify the relationship between the lake colors and the water quality indicators. To pinpoint the causes of the color distinction between the karst lakes in Jiuzhaigou and the plateau freshwater lakes, we conducted principal component analysis (PCA) on the aforesaid two types of lakes, investigated the major indicators that caused the difference in water quality, and compared the difference with One Way ANOVA as well as the box plot especially drawn. Finally, the tristimulus values R, G, B collected at the same depth at the two types of lakes were calculated via equations (4)–(9) and comparison was drawn between the color difference with the naked eye to further expound on the influence of water quality changes over lake colors.

## 3. Results and Discussion

### 3.1. Spectral Reflection Characteristics of the Jiuzhaigou Karst Lakes

To demonstrate the distinctive color characteristics of the karst lakes in Jiuzhaigou, we collected the water-leaving radiance and incident irradiance data at the sampling points of various lakes within the visible wavelength between 380 nm and 780 nm by utilizing the hyperspectral instruments. Figure 3 shows the spectral curves of the normalized water-leaving radiance ($L_{WN}$) data collected at different lakes, from which it can be told that all the curves manifest characteristics of Type II waters such as inland lakes. At the wavelength of 550–580 nm, salient reflection peaks were observed because of the weak absorption of chlorophyll as well as carotene and the scattering of cells. In the Jiuzhaigou karst lakes, due to the lush algae at the bottom, the reflection peaks are more obvious than those in the plateau freshwater lakes. At the wavelength of 620–630 nm, a wave trough was seen in the curves owing to the absorption of light by phycocyanobilin and phycocyanin. At the wavelength of around 670 nm, the chlorophyll contained in the algae has intense absorption towards the red light, hence the noticeable absorption troughs appear in the shallow areas. Whereas in the deep areas, no such troughs were seen as the incident light already started to fade away before it reached the algae at the bottom of lakes. The reflection peaks at the wavelength of 700 nm were formed because absorption by chlorophyll reached the minimum level, while its existence is important proof for algae in the

water. It can be seen from Figure 3 that spectral reflection peaks were formed at CL$_2$; CL$_7$ of Colorful Lake; and ML$_1$, ML$_8$, ML$_9$, and ML$_{11}$ of Mirror Lake as the two lakes are shallow and the chlorophyll contained in the algae at the lake bottom has strong reflection as well as the scattering of red light.

Apart from the typical Type II water spectral characteristics, at the sampling points of part of the relatively deep areas in Colorful Lake (Figure 3a), Mirror Lake (Figure 3b) and Rhinoceros Lake (Figure 3c), the reflective wavelength of the normalized water-leaving radiance is mainly at the blue and green light wavelength (420–570 nm). When the wavelength gets longer than 600 nm, the water-leaving radiance value starts to drop dramatically. Reflection and scattering of visible light by the water of the karst lakes is highly selective for the wavelengths. While at the plateau freshwater lakes of Xiarucuo Lake and Cuoqiong Lake, the spectral reflection curves only present the general characteristics of the inland lakes and there is a noticeable difference between the spectral reflection characteristics at various sampling points. With little algae in the lakes, there are no salient reflection peaks and valleys.

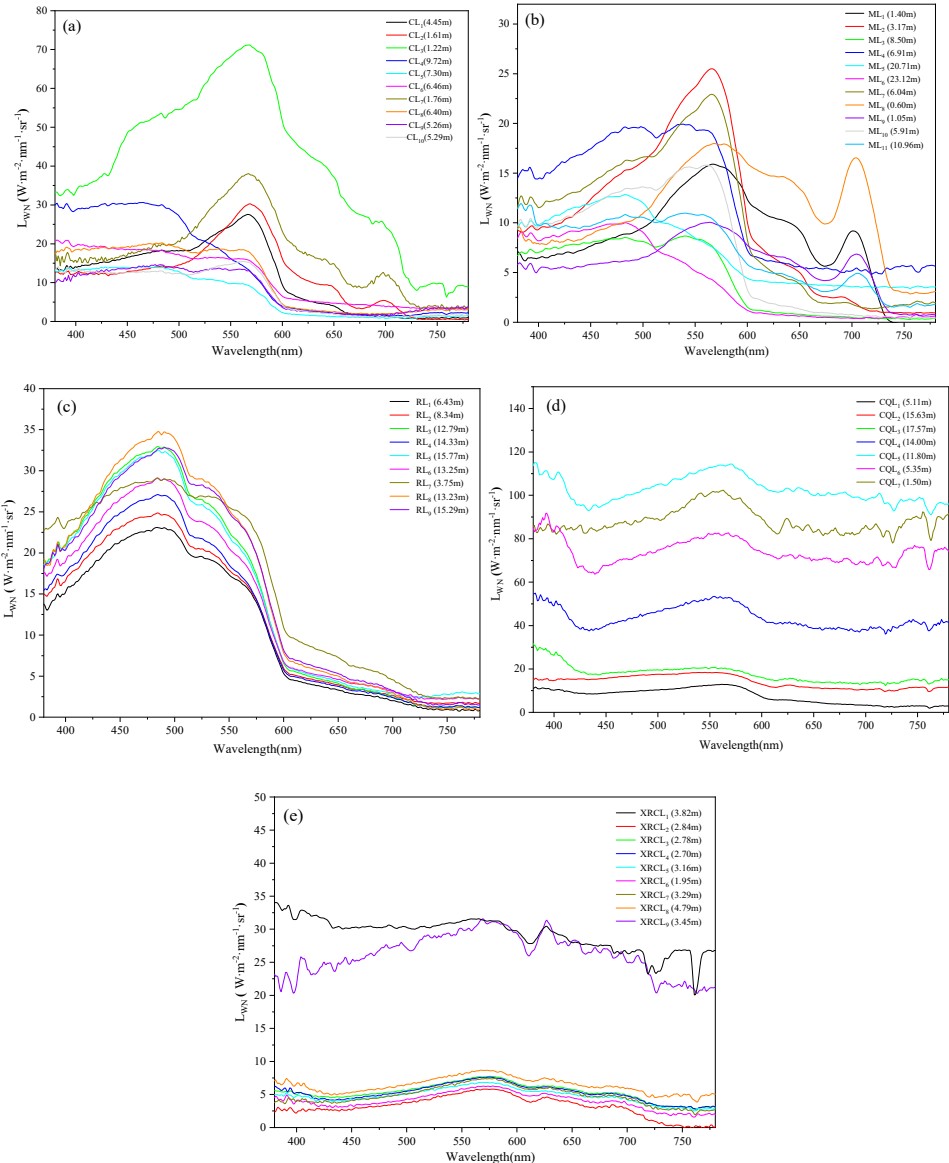

**Figure 3.** Normalized water-leaving radiance curves of (**a**) Colorful Lake; (**b**) Mirror Lake; (**c**) Rhinoceros Lake; (**d**) Cuoqiong Lake; (**e**) Xiarucuo Lake. Curves in different colors stand for the normalized water-leaving radiance data measured at various sampling points of the lakes within the scope of the visible light.

### 3.2. Color Characteristics of the Jiuzhaigou Karst Lakes

We collected the incident irradiance $E_d(R)$, $E_d(G)$ and $E_d(B)$ at the red, green and blue wavelengths measured at the sampling points and normalized the water-leaving radiance $L_{WN}(R)$, $L_{WN}(G)$ and $L_{WN}(B)$, after which the relative proportion of red, green and blue light in the water incident light as well as the reflected light were calculated. From Figure 4, it can be told that the average proportions of red, green and blue light in the incident radiance of the Jiuzhaigou karst lakes are 30.2%, 36.6% and 33.2% respectively, while those of the plateau freshwater lakes are 29.5%, 35.3% and 35.2% respectively. Short-wavelengths in the incident light entering both types of lakes account for comparatively high proportions. On the plateaus, the air is thin and the sky is clear. The proportion of the short-wavelengths received in the lakes during summer rises significantly.

With unique water quality conditions, proportions of the blue and green light reflected by the water of the karst lakes in Jiuzhaigou are 44.5% and 46.8% respectively. The red light reflected is merely 8.7% with long wavelengths. However, proportions of reflection towards the aforementioned three kinds of light at different lakes vary. Reflectivity of red, green and blue light at Colorful Lake is 8.6%, 47.8% and 43.6% respectively. That of green light at Mirror Lake is 48.1%, and of red and blue light is 11.3% and 40.6% respectively. Reflectivity of blue light at the Rhinoceros Lake is as high as 50.4%, while that of red and green light is 5.6% and 44.0%, respectively. Reflectivity of blue light at Rhinoceros Lake is higher than that at Colorful Lake and Mirror Lake. Reflectivity of green light at Mirror Lake is the highest among the three lakes. While at the plateau freshwater lakes, reflectivity of red light with long-wavelengths increases to 27.7%, and that of green and blue light is 41.0% and 31.3% respectively. According to the theory of the Three Primary Colors, various colors can be generated by mixing red, green and blue per different proportions, while combinations of the three colors in the reflected light can reflect the colors of objects seen with the naked eye [58]. Such strong selectivity of wavelengths in the reflection as well as scattering of blue and green light by the lake water is likely to be the direct cause of the blue color of the karst lakes in Jiuzhaigou.

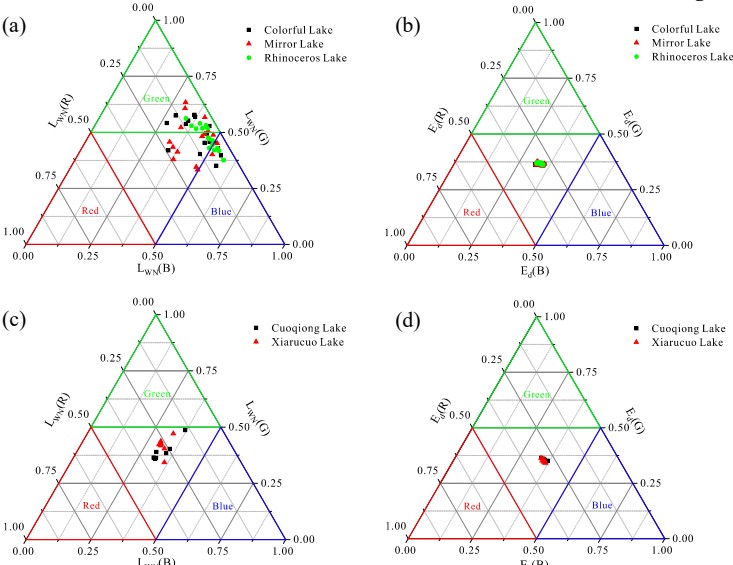

**Figure 4.** Ternary diagram of (**a**) Normalized water-leaving radiance of Jiuzhaigou karst lakes; (**b**) Incident irradiance of Jiuzhaigou karst lakes; (**c**) Normalized water-leaving radiance of plateau freshwater lakes; (**d**) Incident irradiance of plateau freshwater lakes.

### 3.3. Correlation Analysis on Color of the Blue Karst Lakes and the Water Quality Parameters

To delve into the reasons for the selective reflection and scattering of short-wavelengths by lake water, we carried out an analysis on the correlation between the color of the karst lakes in Jiuzhaigou and the water quality indicators. Redundancy analysis was conducted with the tristimulus values R and B as dependent variables and the water quality parameters measured as the explanatory

variables to create an RDA ordination graph. From the RDA analysis results, it can be seen that there is a strong correlation between transparency, TDS and lake colors ($p < 0.01$), with the contribution rates being 69.4% and 8.9% respectively. Correlation between $SI_C$ and lake colors is noticeable ($p < 0.05$), with the contribution rates being 5.5%. The R value is negatively correlated with depth, transparency and positively correlated with TP, CDOM, N/P, and chromaticity. The B value is positively correlated with depth, transparency, $NO_3^-$, and $SO_4^{2-}$ and negatively correlated with $SI_C$, TN, N/P, $Ca^{2+}$, $HCO_3^-$, conductivity, salinity, and TDS, with the correlation N/P > $SI_C$ > TDS > conductivity > salinity > TN > $HCO_3^-$ > $Ca^{2+}$. The $SI_C$ is positively correlated with $Ca^{2+}$, $HCO_3^-$, $Mg^{2+}$, pH, DO, WT, conductivity, salinity, and TDS, and negatively correlated with $NO_3^-$ and $SO_4^{2-}$.

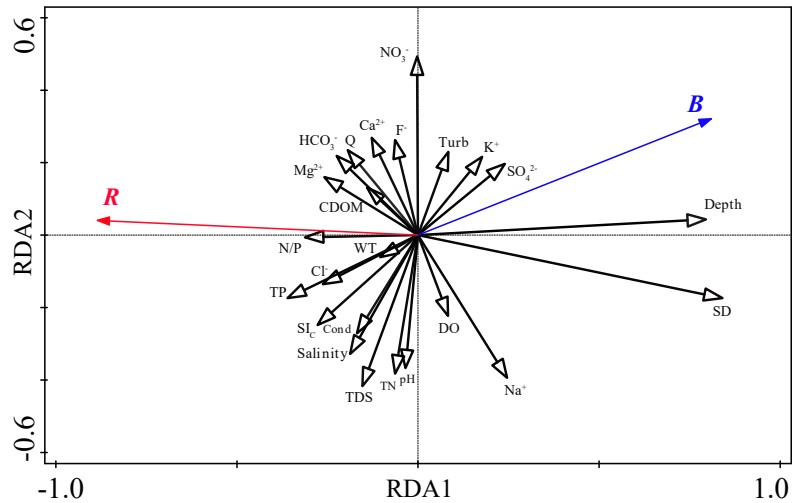

**Figure 5.** RDA ordination graph of karst lake color and water quality parameters.

In the ordination graph, G value is constant 255 and is therefore not displayed. In Figure 5, R and B values of the sampling points are shown with red and blue full lines respectively; water quality parameters are shown with black full lines. Lengths of the arrows stand for impact by various water quality parameters on the lake color. The longer the arrows, the bigger the impact, and vice versa; cosine of the angle between the directions of the arrows and those of the lake color values represents relevance between the two. When the angle is smaller than 90°, the two are positively correlated. When it is bigger, they are negatively correlated. If it is 90°, they are irrelevant.

Next, a correlation heat map was created after the Pearson Correlation Analysis was conducted on the tristimulus values R and B, and the lake water quality parameters of the karst lakes in Jiuzhaigou. As can be seen from Figure 6, the R value is positively correlated with N/P (n = 51, $p < 0.01$), CDOM and chromaticity, and significantly negatively correlated with depth and transparency (n = 51, $p < 0.01$). The B value is significantly positively correlated with depth and transparency (n = 51, $p < 0.01$), positively correlated with $NO_3^-$ and $SO_4^{2-}$; significantly negatively correlated with N/P (n = 51, $p < 0.01$); and negatively correlated with $SI_C$ (n = 51, $p < 0.05$), $Ca^{2+}$, $HCO_3^-$, conductivity, salinity, and TDS. The increase in the depth and transparency of the lake can enable visible light with long-wavelengths to be absorbed by the water selectively. However, the eutrophication of lakes can absorb short-wavelength light in visible light and increase the reflection intensity of long-wavelength light. The $SI_C$ is significantly positively correlated with pH and DO (n = 51, $p < 0.01$), and positively correlated with WT and $Mg^{2+}$ (n = 51, $p < 0.05$). Turbidity is significantly negatively correlated with conductivity, salinity, and TDS (n = 51, $p < 0.01$), and negatively correlated with $HCO_3^-$. The karst lakes in Jiuzhaigou are at a low level of eutrophication and the water is highly clean, which lays the foundation for them to appear bright sky-blue. The abundant $Ca^{2+}$ and $HCO_3^-$ in the Jiuzhaigou karst lakes provide the solid substrate for travertine deposition. When external conditions like pH and water temperature rise, the aqueous solution becomes supersaturated and precipitates calcium carbonate solid, causing the $SI_C$, $Ca^{2+}$、$HCO_3^-$, salinity, TDS, and conductivity in the water to drop,

and concentrations of $SO_4^{2-}$ and $NO_3^-$ to go up comparatively as inhibitors of travertine deposition. Suspended matters of the calcium carbonate scatter and reflect blue and green light and other short-wavelengths selectively, which make the proportions of blue and green light received by human eyes increase and those of red light and other long-wavelengths decrease. At this point, the lakes are blue in human eyes. In the correlation heat map, two clusters are divided based on the lake color tristimulus values. Various water quality factors affect the lake color by impacting the ratios of reflected light. pH, WT, DO, TN, TP, turbidity, depth, and transparency are closely related to short-wavelength light reflected from lakes. Nevertheless, water quality factors such as $SI_C$, $K^+$, $Mg^{2+}$, $Ca^{2+}$, $HCO_3^-$, $NO_3^-$, $SO_4^{2-}$, N/P, CDOM, salinity, TDS, and conductivity participate in the reflection and scattering process of long-wavelength light in lakes. This indicates that the travertine deposition process can reduce the reflection intensity of long-wavelength light and selectively increase the reflection intensity of short-wavelength light. Water quality factors related to travertine deposition, and eutrophic water quality factors such as TN, TP, and CDOM may also be related to the color formation mechanism of the blue karst lakes.

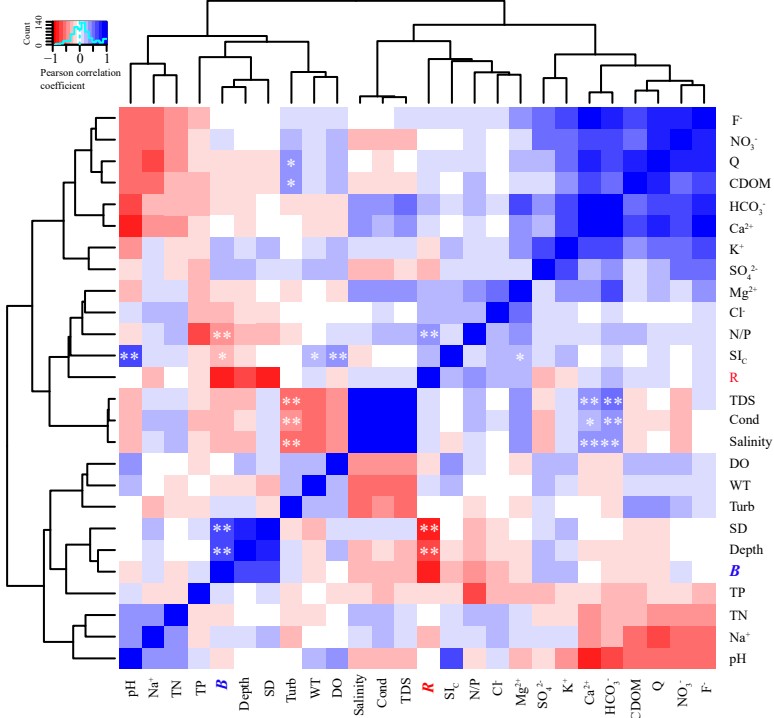

**Figure 6.** Pearson correlation heat map based on lake colors and water quality parameters. ** Correlation is significant at the 0.01 level; * Correlation is significant at the 0.05 level. In the correlation heat map, on the X and Y axes are the lake color R value, G value and water quality parameters. The Pearson Correlation Coefficients (r) are demonstrated in different colors, with blue standing for positive correlation and red negative correlation. The deeper the color, the higher the correlation.

Through the above analysis, it can be told that travertine deposition is the core process for the formation of the blue karst lakes in Jiuzhaigou, while the selective reflection and scattering by the calc-sinter particulate matters towards light with short-wavelengths such as blue and green light is the optical basis for the formation of the blue karst lakes. Moreover, water quality factors can indirectly alter lake color by affecting the travertine deposition process, while the depth, transparency and eutrophication process have a direct influence over changes to the color of the blue lakes.

Multiple factors are influential for the formation of the blue color of the karst lakes, chief among which are lake depth and transparency. According to research, due to selective absorption, pure water can realize the maximum absorption at 750–760 nm and the minimum at the short- wavelength

of around 418 nm. In the visible light spectrum, pure water has the highest transmission rate of blue light and the highest attenuation rate of red light [59,60]. On account of the coprecipitation between phosphate and calcium carbonate, the process of travertine deposition at the karst lakes has a fixation effect on nutrients such as phosphate. There are dissolution and precipitation in the colloidal solution formed between calcium carbonate and water, which can effectively absorb the suspended impurities and purify the water [61–63]. Besides, the crown density of the vegetation along the banks of Jiuzhaigou Natural Reserve is high, there are few surface runoffs and the sand concentration in the water is low, which makes the karst lakes extraordinarily clear. When the depth of the lake reaches a certain level, the incident light has a substantial depth of transmission, and there is intense selective absorption by the water towards long-wavelengths such as red light. As a result, the water primarily reflects blue and green light. At this point, the deeper the lake, the bluer the water. As we collected samples from lakes, DJI Mavic Pro 2 was utilized to film the real-time color of Colorful Lake. With the water-leaving radiance data measured in situ, lake color at five sampling points was simulated (Figure 7). According to the results, depth is decisive to the blue color of the Lake. In the shallow areas of Colorful Lake, the color is affected by the substrate. At Sampling Point No.2 (CL2), algae are flourishing, which makes the lake menthol. Due to the weak absorption of chlorophyll as well as carotene and the scattering of cells in the algae, the reflection peak formed at 550–580 nm makes the water-leaving radiance value at the blue light wavelengths relatively high. Sampling Point No.3 (CL3) is close to the inlet of the lake, where the depth is merely 1.22 m. The algae and sand deposited at the bottom of the lake make the water yellow-green. In the deep area, the lake color becomes blue and gets deeper. As the depths of Sampling Points No.9 (CL9) and No.10 (CL10) are smaller than those at Sampling Point No.5 (CL5), the ratios of reflection and scattering of blue light are also smaller, which directly lead to the changes to the lake color. From CL9 to CL10 and CL5, the lake color gradually changes from electric blue to celeste and turquoise blue.

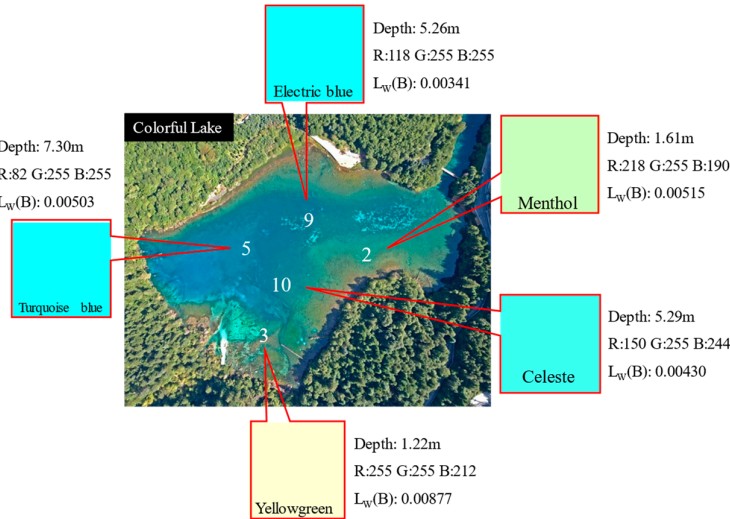

**Figure 7.** Real-time color at the sampling points of Colorful Lake (the picture was taken in August 11, 2019.).

Influence by depth and transparency over the blue color of the lakes also explains why some lakes in Jiuzhaigou without noticeable travertine deposition and some plateau freshwater lakes appear blue as well. For instance, the rate of travertine deposition at Upper Season Lake, Lower Season Lake and Long Lake in Jiuzhaigou is relatively low, but the lake water mainly comes from precipitation and underground karst spring, hence very few suspended matters. When the water reaches a certain level of depth, with the long-wavelengths being selectively absorbed, the lakes tend to be blue. In some plateau freshwater lakes, due to the thin air and clear sky, transmission of the short-wavelengths in the incident light emitted by the sun is weak. With the altitude increases, the proportion of the short-wavelengths in the incident light received by the lake water rises significantly,

and the lake water primarily reflects the short-wavelengths. In addition, as the lakes are seldom affected by human activities, the level of eutrophication is low and the lake water is pure. Therefore, the depth of the water becomes the primary determinant of the lake color.

Besides geographical conditions such as depth and transparency, some water quality factors can also influence travertine deposition and therefore indirectly alter the lake colors. Phosphate compounds, $NO_3^-$ and $SO_4^{2-}$, are deemed as inhibitors for calcite deposition and have a substantial inhibitory effect on travertine deposition. Phosphate can be absorbed by the surface of the calcite crystal and inhibit travertine deposition by hindering the formation of the active crystal [64,65]. The absorption of phosphate by calcite and the inhibitory effect by phosphate on calcite deposition is closely related to the pH of the solution and its supersaturation [66–69]. Under certain temperature and salinity, elevating the pH can significantly increase the content of $CO_3^{2-}$ in the solution, thus consolidating the ion activity product of calcium carbonate in the water and creating enabling conditions for the formation of deposits. A solution with supersaturation is more likely to be affected by the retardation of phosphate. Research has found that sulphate can slow down the nucleation crystallization of calcium carbonate and alter the crystal shapes as well as surface property. When concentrations of sulphate and nitrate in the solution lower, the calcite saturation index ($SI_C$) rises, contributing to the deposition process [38,70,71]. Except for inorganic ions, it is generally believed that dissolved organic matter (DOM) or dissolved organic carbon (DOC) is essential to travertine deposition. Lebron et al. found that the existence of DOC would reduce the sizes of the calcite crystal matters. When concentrations of the carbon absorbed by the calcite crystal reach 0.11mmol/ $m^2$, no calcite deposits will be found [72]. Zhang et al. found the existence of DOM greatly changed the shapes of the calcium carbonate crystal. Without DOM, the calcium carbonate crystal formed is regularly rhombus or spherical-shaped and the surface is smooth. When there is DOM, however, as it occupies the locus of crystal growth, the growth process is interrupted. Consequently, the surface of the crystal is rough and there are flaws [73]. The inhibitory effect of organisms in the calcium carbonate deposits is relevant with the formation of calcium complex, after which concentrations of free calcium will be lowered, thus reducing the saturation of calcium carbonate and impeding calcium carbonate deposition [74]. Colored dissolved organic matters (CDOM) are one type of dissolved organism, and it is speculated that they might serve as an inhibitor for travertine deposition as well. Water quality factors can indirectly alter the lake colors by influencing the travertine deposition process. In Jiuzhaigou, the content of phosphorus in the lakes is low and there are few surface runoffs. Consequently, the lakes become oligotrophic and the content of $NO_3^-$, $SO_4^{2-}$ and organic matters is low, hence the insignificant inhibition of water quality factors against travertine deposition. Inversely, the low concentrations of phosphorus and organic matters in the water are conducive to the formation of calcium carbonate matters, while the concentrations of $NO_3^-$ and $SO_4^{2-}$ that have retardation against the rate of travertine deposition rise relatively because of the travertine deposition process. In the karst areas of Jiuzhaigou, the ample $Ca^{2+}$ and $HCO_3^-$ widely existing in the lakes provide a solid foundation for travertine deposition. The increase of $Na^+$, $K^+$ and $F^-$ that are related to travertine deposition is beneficial to the deposition process. When the temperature and pH value of the karst lakes rise, the solution becomes supersaturated and therefore gives rise to calcium carbonate deposits. Affected by travertine deposition ($SI_C > 0$), $Ca^{2+}$ and $HCO_3^-$ drop with the altitude.

Furthermore, the degree of eutrophication can directly change the lake colors. Also dubbed as the "Yellow Matter", CDOM is a type of important light absorption matters that mostly originates from the inflow of runoffs and creature activities, which can absorb the short-wavelengths such as ultraviolet and blue light in the spectrum to a large extent. While the accumulation and degradation of planktonic algae is also a major source of CDOM in the water [75–78]. The absorption coefficient of CDOM is closely intertwined with the nutritional status of the water, which tends to be high in the eutrophicated lakes. When the organic matters and the content of N and P nutrients increase, plankton will thrive, resulting in eutrophication. At the same time, the residue of the dead algae can also release CDOM, thus changing the light absorption characteristics of the water, darkening the water and reducing the transparency. Phosphorus is the leading restricting component in the karst lakes in Jiuzhaigou, where the comparatively low CDOM and the ratio of nitrogen to phosphorus

reduce the reflection of long-wavelengths such as red light by the lake water. That creates favorable conditions for the formation of the blue color of the lakes.

### 3.4. Karst Lake Water Quality-Color Equation

It can be told from the correlation matrix formed by the R value, B value and water quality that they are linearly correlated. After that, stepwise linear regression analysis was conducted between the water quality indicators related to the R value such as depth, transparency, $F^-$, $NO_3^-$, $SO_4^{2-}$, $SI_C$, conductivity, TDS, salinity, N/P and CDOM, and the R value coupled with the abovementioned RDA and correlation cluster analysis results; meanwhile, it was also conducted between the water quality indicators related to the B value such as $SI_C$, $Na^+$, $HCO_3^-$, $Ca^{2+}$, pH, WT, DO, TN, TP, depth, and transparency. An equation set was deduced as follows:

$$R = -15.49SD + 2.13TDS + 164.74SI_C - 368 \qquad R^2 = 0.568, P < 0.01, N = 51$$

$$B = 4.77SD + 199.38K^+ - 76.71SI_C - 5.93DO - 0.67Cond + 483 \qquad R^2 = 0.537, P < 0.01, N = 51$$

Where SD represents transparency of the Jiuzhaigou karst lakes (m), $K^+$ is content of potassium ion, $SI_C$ is calcite saturation index, TDS is total dissolved solids (ppt), DO is dissolved oxygen (mg/L), and Cond is conductivity ($\mu s\ cm^{-1}$).

It can be seen from the equation set that transparency is essential to the intensity of reflection of red and green light by lakes. Travertine deposition can reduce the intensity of long-wavelengths such as red light and stimulates selective reflection as well as the scattering of short-wavelengths such as blue light. Moreover, components related to travertine deposition like $K^+$, conductivity and total dissolved solids, and eutrophication indicators such as DO, are also central to the formation of the lake colors.

### 3.5. Color Distinction between the Jiuzhaigou Karst Lakes and the Plateau Freshwater Lakes

### 3.5.1. Comparison between Water Quality of the Jiuzhaigou Karst Lakes and the Plateau Freshwater Lakes

The Principal Component Analysis was carried out on the water quality data of the Jiuzhaigou karst lakes and the plateau freshwater lakes to assess the water quality parameter difference between the two. From Figure 8, the variance contribution rate of the first principal component (PC1) is 61.9% and that of the second (PC2) is 9.4%. It can be seen from the 95% confidence ellipse that there is a clear distinction between water quality characteristics of the karst lakes and the plateau freshwater ones (Figure 8b), and water quality parameters that affect each lake vary. The primary hydro-chemical indicators closely linked to PC1 are TDS, conductivity, $K^+$, salinity, $SO_4^{2-}$, $Na^+$, $HCO_3^-$, $Ca^{2+}$, $F^-$, and $SI_C$, with the loadings being -0.2456, -0.2455, -0.2451, -0.2442, -0.2440, -0.2419, -0.2417, -0.2412, -0.2300, and -0.2296 respectively. Next to those are chromaticity, $NO_3^-$, $Mg^{2+}$, CDOM, DO, turbidity, pH, WT, and TN. Indicators closely related to PC2 are depth, pH, transparency, N/P, WT, CDOM, turbidity, chromaticity, $F^-$, and $SI_C$, with the factor loadings being 0.4231, 0.3830, 0.3572, -0.2842, -0.2668, -0.2469, -0.2316, -0.2152, -0.1882, and 0.1838 respectively. Next to those are $NO_3^-$, TP, $HCO_3^-$, $Ca^{2+}$, $K^+$, $SO_4^{2-}$, salinity, TDS, conductivity, elevation, $Mg^{2+}$, $Na^+$, TN, $Cl^-$ etc. Among all the water quality indicators, those that are relevant to travertine deposition like $SI_C$, conductivity, TDS, salinity, $Na^+$, $K^+$, $Ca^{2+}$, $HCO_3^-$, and pH value have the highest contribution rates to the water quality characteristics of the karst lakes. Besides, $NO_3^-$, N/P, TN, etc. can also manifest the water quality of the karst lakes. In contrast, the water quality of the plateau freshwater lakes is mainly affected by chromaticity, turbidity and CDOM.

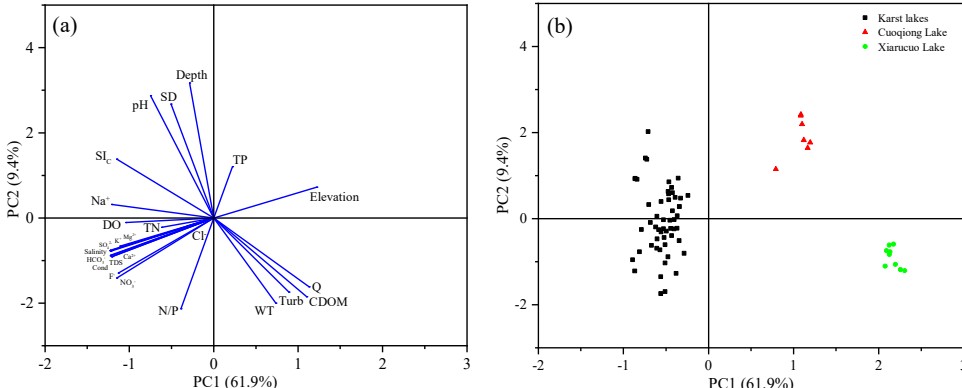

**Figure 8.** Principal component analysis on the lake water quality parameters. (**a**) Loading data of factors; (**b**) Scores of the sampling points. The ellipse shown in the figure is the 95% confidence ellipse.

The analysis was conducted on the major indicators that lead to the difference between the water quality of the karst lakes in Jiuzhaigou and the plateau freshwater lakes via the one-way ANOVA test, including $Ca^{2+}$, $HCO_3^-$, conductivity, TDS, salinity, $SI_C$, TP, CDOM, and transparency. A box plot showing different water quality was created. According to Figure 9, there is a significant distinction ($p < 0.05$) between the karst lakes in Jiuzhaigou and the plateau freshwater ones in terms of the water quality indicators relevant to travertine deposition such as $Ca^{2+}$, $HCO_3^-$, electrical conductivity, TDS, and salinity. Besides, those indicators tend to decrease with the altitude at the former. The content at different sampling points varies substantially, which is related to the distinctive hydrological characteristics of the karst lakes [79]. Whereas the water quality of the various sampling points at the karst lakes in Jiuzhaigou differs greatly, that at the freshwater lakes is more similar. There is no clear difference in the calcite saturation index among the three karst lakes, namely, the rate of travertine deposition at those lakes is not much different. Compared with those in the karst lakes in Jiuzhaigou, concentrations of CDOM in the plateau freshwater lakes are a lot higher ($p < 0$ .05). The transparency is relatively low, and yet the content of TP is alike. The plateau freshwater lakes surveyed are distant from areas with human activities and there is no obvious eutrophication. However, crown density of the vegetation along the banks of lakes is low, which is likely to be related to the surface runoffs surrounding the lakes: the mud and sand brought by precipitation and the inflow of organic matters result in the increase of suspended matters in the lakes as well as a great number of CDOM, which makes the lakes primarily dark yellow even at the same depth. Although the Cuoqiong Lake is comparatively transparent, with insignificant travertine deposition, the lake color is mainly affected by the color of the substrate. There is no significant distinction of the TP content between different lakes, and the content of phosphorus is relatively low, whereas phosphorus is the leading limiting component of eutrophication at the two types of lakes.

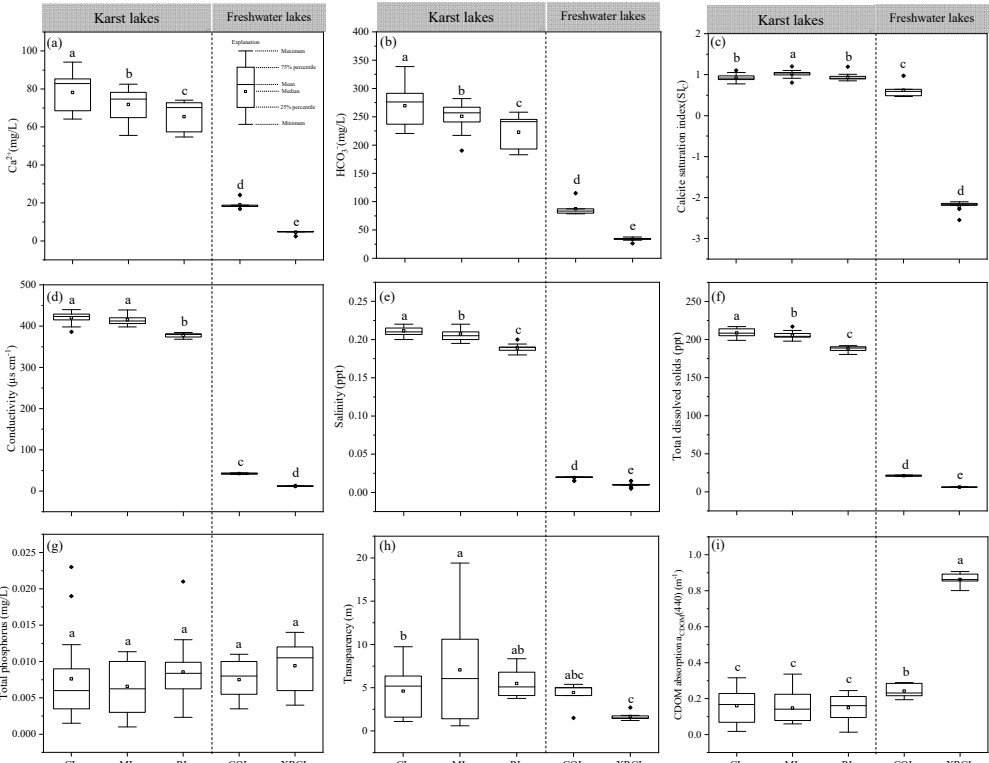

**Figure 9.** Box plot of water quality parameters for different lakes. (**a**) Calcium ion; (**b**) Bicarbonate ion; (**c**) Calcite saturation index; (**d**) Conductivity; (**e**) Salinity; (**f**) TDS; (**g**) TP; (**h**) CDOM concentrations; (**i**) Transparency. CL represents Colorful Lake, ML is Mirror Lake, RL is Rhinoceros Lake, CQL is Cuoqiong Lake, XRCL is Xiarucuo Lake.

### 3.5.2. Comparison of Colors of the Jiuzhaigou Karst Lakes and the Plateau Freshwater Lakes

To further investigate changes to colors of the Jiuzhaigou karst lakes and the plateau freshwater lakes caused by different water quality, equations (4)–(9) were used to calculate the real-time colors at sampling points $CL_8$, $ML_{10}$, $RL_1$, $CQL_6$, and $XRCL_8$ of the same depth based on the water-leaving radiance data. At the similar depths, there is a noticeable color difference between the karst lakes in Jiuzhaigou and the plateau freshwater lakes (Table 1); the karst lakes are primarily blue and green and the water has relatively high water-leaving radiance value in the blue light band, meaning the lakes mainly reflect and scatter blue light. While at the plateau freshwater lakes, with high turbidity and insignificant travertine deposition, the lake colors are largely affected by the substrate and therefore not blue and green as the karst ones.

**Table 1.** Color Differences at Different Sampling Points.

| Sites | $CL_8$ | $ML_{10}$ | $RL_1$ | $CQL_6$ | $XRCL_8$ |
|---|---|---|---|---|---|
| Depth (m) | 6.40 | 5.91 | 6.43 | 5.35 | 4.79 |
| Lake color | Electric Blue | Celeste | Electric Blue | Ivory White | Yellow Green |
| (R, G, B) | (116, 255, 255) | (113, 255, 237) | (118, 255, 255) | (255, 255, 241) | (255, 255, 210) |

## 4. Conclusions

The karst lakes in Jiuzhaigou Natural Reserve are an important type of karst landscape, and yet there is little research on the formation mechanism of its blue color. Via in situ field collection of hyperspectral data and indoor water quality analyzing experiment, this paper delves into the formation of the blue color of the typical karst lakes in Jiuzhaigou, which found that travertine deposition lies in the core of the formation process. Apart from that, lake depth, transparency, water quality parameters that affect travertine deposition, and the eutrophication process are also central to the formation of the lake colors. Travertine deposition and distinction in the CDOM content are the primary reasons that lead to the color differences between the Jiuzhaigou karst lakes and the plateau freshwater lakes. Nonetheless, the specific optical process of selective scattering by calcium carbonate suspended matters towards visible light remains unclear and further research is required on the difference in the formation mechanism of the blue karst lakes in different seasons.

**Author Contributions:** G.S. conceived and designed the experiments; X.L. and D.Z. performed the experiments; X.L. analyzed the data; G.S. and A.P.-M. reviewed drafts of the paper; M.Z. and M.S assisted in water sampling and experiments; W.X and J.D assisted in sampling and field test. All authors have read and agreed to the published version of the manuscript.

**Funding:** This research was funded by the Second Tibetan Plateau Scientific Expedition and Research (STEP) Program (2019QZKK0302), the National Key Research and Development Program of China (2016YFC0501803), Sichuan Science & Technology Bureau (2017SZ0080、2017HH0084、2017NFP0223、2018SZ0329、2018HH0021 、2018HH0008、2018NFP0107、2018NZ0080、2018NZ0078、2019YFH0042、2019YFH0132、2019YFS0468、 20GJHZ0155 、 20GJHZ0085 、 20ZDYF1645 、 20YZTG0054 、 20FPCY0285) and Jiuzhaigou Post-Disaster Restoration and Reconstruction Program Research on Restoration and Protection of World Natural Heritage.

**Acknowledgments:** We would like to convey our thanks to the Jiuzhaigou Nature Reserve Administrative Bureau for their support during the field water sampling process. Our gratitude also goes to the Public Experimental Technological Center of the Chengdu Institute of Biology, Chinese Academy of Sciences for its help with the sample test.

**Conflicts of Interest:** The authors declare no conflict of interest.

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
