# Peer review of "The Color Formation Mechanism of the Blue Karst Lakes in Jiuzhaigou Nature Reserve, Sichuan, China"

_water, doi:10.3390/w12030771_

Round 1

Reviewer 1 Report

This manuscript tries to explore the color formation mechanism of the lakes in Jiuzhaigou Nature Reserve, which is a interesting and meaningful work. But I suggest some minor revisions before publishing it.

(1) Figure 1. The map is not clear, and readers can not see all of the lakes concerned.

(2) Page 6 Section 2.6. Water-leaving radiance, not the normalized water-leaving radiance, is used to calculate tristimulus values. Will the sunlight or skylight affect the RGB color?

(3) Figure 3. Obvious noise is found in spectral regions like 380-400 and 750-780, will they affect the results .

Author Response

Dear Reviewer:

       Thank you for your comments concerning our manuscript entitled “The Color Formation Mechanism of the Blue Karst Lakes in Jiuzhaigou Nature Reserve, Sichuan, China” (ID: water-730019). Those comments are all valuable and very helpful for revising and improving our paper, as well as the important guiding significance to our research. We have studied comments carefully and have made corrections which we hope to meet with your requirements for a publication. The main corrections in the paper and the response to your comments were uploaded as a Word file. Please see the attachment.

       We appreciate your warm work earnestly and hope that the correction will meet with approval.

       Once again, thank you very much for your comments and suggestions. If you have any queries, please don’t hesitate to contact me at the address below.

       Thank you and best regards.

       Yours sincerely,

       Geng Sun

       Chengdu Institute of Biology, Chinese Academy of Sciences

       E-mail:sungeng@cib.ac.cn

Reviewer 2 Report

General comments

The paper entitled “The Color Formation Mechanism of the Blue Karst Lakes in Jiuzhaigou Nature Reserve, Sichuan, China” treats about a topic of the highest interest in the scope of water resources safeguarding and sustainability. The used language is appropriate and fluent. The typographical outline is effective and satisfactory. KeyWords are pertinent to paper content and appropriate. The References section is rich both in quantity and quality and the effort to make an effective review of the state of the art about predictive methods is commendable. Graphic representations are all of good quality and self-explaining. The topic falls under the journal scope without any doubt.

The methods applied for studying the formation process of Blue Karst Lakes, namely PCA, multivariate analysis etc. are not novel but, in the opinion of this reviewer, the interest of the present study concerns the application of colorimetrical methods to water quality assessment. Data pre-treatment is well conducted for avoiding interfering effects (e.g. from atmosphere) that can impact adversely on the next statistical stage. Statistical modelling is appropriate, correctly conducted and effective. The conclusions are well balanced with respect the findings. In conclusion, it is opinion of this reviewer that the present work can be accepted for publication so it is.

Author Response

Dear  Reviewer:

     Thank you for your comments concerning our manuscript entitled “The Color Formation Mechanism of the Blue Karst Lakes in Jiuzhaigou Nature Reserve, Sichuan, China” (ID: water-730019). We deeply appreciate your encouragement and valuable comments. These comments helped us improve our manuscript and provided important guidance for future research. In future research, we will consider applying appropriate analytical methods and further improve the level of research.

     Once again, thank you very much for your comments and suggestions. If you have any queries, please don’t hesitate to contact me at the address below.

     Thank you and best regards.

     Yours sincerely,

     Geng Sun

     Chengdu Institute of Biology, Chinese Academy of Sciences

     E-mail: sungeng@cib.ac.cn